# Comparative Transcriptomic Analysis of Temozolomide Resistant Primary GBM Stem-Like Cells and Recurrent GBM Identifies Up-Regulation of the Carbonic Anhydrase *CA2* Gene as Resistance Factor

**DOI:** 10.3390/cancers11070921

**Published:** 2019-06-30

**Authors:** Ricarda Hannen, Martin Selmansberger, Maria Hauswald, Axel Pagenstecher, Andrea Nist, Thorsten Stiewe, Till Acker, Barbara Carl, Christopher Nimsky, Jörg Walter Bartsch

**Affiliations:** 1Department of Neurosurgery, UKGM, Philipps University Marburg, Baldingerstraße, 35033 Marburg, Germany; 2Department of Radiation Cytogenetics, Helmholtz Zentrum München, German Research Center for Environmental Health, Ingolstaedter Landstr. 1, 85764 Neuherberg, Germany; 3Department of Neuropathology, UKGM, Philipps University Marburg, Baldingerstraße, 35033 Marburg, Germany; 4Genomics Core Facility, Philipps University Marburg, Hans-Meerwein-Straße 3, 35043 Marburg, Germany; 5Institute of Molecular Oncology, member of the German Center for Lung Research (DZL), Philipps University Marburg, Hans-Meerwein-Straße 3, 35043 Marburg, Germany; 6Institute for Neuropathology, Justus-Liebig University Gießen, Arndtstr. 16, 35392 Gießen, Germany

**Keywords:** glioblastoma, GBM Stem-like cells, temozolomide, chemoresistance, GBM recurrence, transcriptomics, acetazolamide, carbonic anhydrase 2

## Abstract

About 95% of patients with Glioblastoma (GBM) show tumor relapse, leaving them with limited therapeutic options as recurrent tumors are most often resistant to the first line chemotherapy standard Temozolomide (TMZ). To identify molecular pathways involved in TMZ resistance, primary GBM Stem-like Cells (GSCs) were isolated, characterized, and selected for TMZ resistance in vitro. Subsequently, RNA sequencing analysis was performed and revealed a total of 49 differentially expressed genes (|log2-fold change| > 0.5 and adjusted *p*-value < 0.1) in TMZ resistant stem-like cells compared to their matched DMSO control cells. Among up-regulated genes, we identified carbonic anhydrase 2 (CA2) as a candidate gene correlated with glioma malignancy and patient survival. Notably, we describe consistent up-regulation of CA2 not only in TMZ resistant GSCs on mRNA and protein level, but also in patient-matched clinical samples of first manifest and recurrent tumors. Co-treatment with the carbonic anhydrase inhibitor Acetazolamid (ACZ) sensitized cells to TMZ induced cell death. Cumulatively, our findings illustrate the potential of CA2 as a chemosensitizing target in recurrent GBM and provide a rationale for a therapy associated inhibition of CA2 to overcome TMZ induced chemoresistance.

## 1. Introduction

Glioblastoma (GBM) is the most lethal brain tumor with a median survival of only 15 months [1]. Despite an aggressive standard of care consisting of surgical resection, adjuvant radiation and chemotherapy with temozolomide (TMZ) [2,3] virtually all GBMs recur [4].

GBM is characterized by a high degree of heterogeneity on phenotypic, genetic and cellular levels [5]. As other solid tumors GBM are composed of various brain resident as well as transformed cell types: there are rapidly multiplying tumor cells which make up the bulk of the tumor mass and, on the other hand, there are self-renewing cell types, often regarded as Glioblastoma stem cells (GSCs) [3,5,6]. Whereas the differentiated tumor cells are eradicated by therapy due to their high proliferation rate, the latter are thought to exert increased resistance to adjuvant therapy and tumor initiating capacity as a source of glioma recurrence [6,7,8,9]. It is well established that GSC driven recurring tumors are resistant to further treatment, but the underlying molecular changes are not fully understood [10,11]. There are several proposed resistance mechanisms such as metabolic inactivation of drugs, inhibition of conversion from prodrug to bioactive drug, increased drug efflux, and increased DNA repair [8]. Importantly, the detailed analysis of these processes and their contribution to GBM resistance promises the discovery of additional targets for combinatorial therapies to overcome resistance to TMZ.

To identify new potential therapeutic targets, we exploited an in vitro approach of GBM recurrence by generating TMZ resistant primary GSCs. Subsequently, TMZ resistant and DMSO control cells were compared by RNA sequencing analyses. Interestingly, only a small number of genes were consistently affected by TMZ treatment when comparing TMZ-resistant GSCs with recurrent GBM patient samples. The most consistently up-regulated gene in TMZ resistant GSCs was the gene encoding Carbonic Anhydrase 2 (CA2). As proof of principle, we identified CA2 overexpression as a characteristic of TMZ resistant GSCs and recurrent TMZ treated GBMs, moreover its inhibition chemosensitized TMZ resistant GSCs.

## 2. Results

### 2.1. Characterization of Primary GBM Stem-Like Cells Compared to the Established Cell Line U87

Fresh tumor tissues from three individual patients with histologically confirmed GBM (Table 1) were processed for cell isolation. Cells were cultured under serum free GSC conditions.

Stemness and GBM specific features were documented in a workflow consisting of in vitro and in vivo experiments. This workflow included stimulation of cells with FCS to show their ability to differentiate. Upon addition of FCS there were phenotypic changes observed as cells did no longer grow in spheres but attached to the cell culture surface in an adherent monolayer (Figure 1A). Additionally, down-regulation of stem cell marker CD133 and induction of GFAP (Glial Fibrillary Acidic Protein) gene expression as differentiation marker were detected in qPCR analysis (Figure 1A). To further substantiate the stem-like phenotype of the cells, a side population analysis was conducted. Here a population of cells with a higher efflux, hence a lower intracellular concentration of Hoechst dye, was identified. Inhibition of ABC transporters with verapamil and concomitant blockage of efflux confirmed specificity of the side population as an efflux was no longer detectable. As exemplified for the primary cells 175 this resulted in a decrease of the side population from 2.56 to 0.68% cells compared to the original population (Figure 1B). The tumor cells were further characterized for GBM specific traits including invasiveness and tumorigenicity. The invasive potential, quantified as the percentage of cells invading along a serum gradient into a synthetic extracellular matrix, was significantly higher for all three primary GBM cells compared to the standard GBM cell line U87 (Figure 1C). These advantages of primary cells were also observed upon stereotactic injection into nude mice and monitoring of tumor growth. While U87 cells led to a defined tumor field that in MRI resembles a meningioma, primary cells generated a highly invasive tumor difficult to visualize by MRI (Figure 1D). The invasive behavior of primary cells was even more obvious in histopathological examination of the tumor. The main portion of cells was found to have invaded along the corpus callosum away from the point of injection. Upon closer inspection the tumor shows many mitoses but also numerous apoptotic cells resembling the histopathological picture of a human GBM as exemplified here by representative images from a tumor caused by the primary cells 175 (Figure 1E). These results confirm that our GSC lines recapitulate typical GBM features both in vitro and in vivo as they accurately reflect the features of GBM as a highly infiltrating tumor.

### 2.2. Long Term Exposure to TMZ Generates Resistant Cells as An In Vitro Model of Reccurence

After establishing a suitable in vitro model of GBM, the further aim was to mimic TMZ resistant recurrence. To this end, GSCs were continuously exposed to 10 µM TMZ or the equivalent amount of DMSO, respectively, over a period of at least 20 weeks, this allowed them to acquire resistance. Dose-response-curves show the acute response of TMZ resistant and DMSO control cells after this time, here IC_50_ values of TMZ resistant cells increased between 15 and 100-fold compared to DMSO control cells (Figure 2). This effect seemed to be due to genetic changes as it persisted even after multiple freeze-thaw cycles.

Notably high TMZ concentrations were necessary in order to achieve a complete reduction of viability of TMZ resistant cells. This led to increased amounts of the toxic solvent DMSO, since the solubility of TMZ is relatively low. The percentage of DMSO for each respective TMZ concentration is given in Table 2.

### 2.3. Identification of Genetic Alterations Causing TMZ Resistance by RNA Sequencing Analysis

RNA sequencing analysis was conducted to uncover genetic alterations causing the TMZ resistance. Principal component analysis (PCA) and subsequent visualization of PC1 and PC2 resulted in a clustering of samples by GSC line. Here TMZ resistant, DMSO control and untreated (ctr) cells from one patient form a cluster, rather than a clustering by treatment conditions of different patients. This was to be expected and serves as a quality control for the acquired data (Appendix B, Figure A1).

Results obtained from *DEseq2* revealed a total number of 49 differentially expressed genes based on the pre-defined thresholds (adjusted *p*-value < 0.1 and |log2-fold change| > 0.5) between TMZ resistant and their paired DMSO control GSCs. These findings are visualized in Figure 3A using a volcano plot and are summarized in Appendix B
Table A1.

Gene Set Enrichment Analysis (GSEA) using the Hallmark gene sets revealed six sets (adjusted *p*-value < 0.05) enriched in TMZ resistant cells: TNFA_SIGNALING_VIA_NFKB, INFLAMMATORY_RESPONSE, HYPOXIA, IL6_JAK_STAT3_ SIGNALING, EPITHELIAL_MESENCHYMAL_TRANSITION, APOPTOSIS. The gene set OXIDATIVE_PHOSPHORYLATION was depleted. Normalized enrichment scores (NES) are visualized for all 50 Hallmark gene sets in Figure 3B.

### 2.4. qPCR Based Validation of Differentially Expressed Genes from RNA Sequencing Points to CA2 as the Most Consistent Mediator of TMZ Resistance

Validation of differential expression results derived from the transcriptomic RNAseq data was carried out by qPCR analysis of 10 up-regulated and 10 down-regulated genes which presented the most homogenous effects in all three individual TMZ-DMSO pairs.

In a first step qPCR analysis was conducted from cell lysates from three independent experiments in triplicates for all 20 genes. Notably, only a single down-regulated gene, *NKAIN1*, and a single up-regulated gene, *CA2*, showed consistent and reproducible results (Figure 4).

Considering a therapeutic application, up-regulation of a gene inducing resistance appears more relevant, since, from a pharmaceutical point of view, inhibiting a gene product is more practicable than inducing its expression. Therefore, we further investigated CA2 but not NKAIN1.

### 2.5. CA2 Overexpression Is Associated with TMZ Resistance in Primary GBM Stem-Like Cells as well as in Patient Matched Tissue Samples of Primary and Recurrent GBMs

Both our RNA sequencing (Figure 5A) as well as the qPCR analysis (Figure 5B) confirmed up-regulation of CA2 in TMZ resistant GSCs identifying CA2 as a potential cause for TMZ resistance. To complement these results, we first inspected if this correlation was also to be observed on the protein level. Indeed, western blot revealed an up-regulation of CA2 in all three TMZ-DMSO pairs (one representative blot shown in Figure 5C). To further substantiate the link between CA2 overexpression and TMZ resistance we then turned to tissue samples of first manifested and corresponding recurrent tumors from the same patient. All eight patients received 6 cycles of TMZ after their initial surgery according to the “Stupp” scheme, rendering the respective tumor tissue a suitable model. Further patient data are summarized in Table 3.

qPCR conducted on mRNA extracted from freshly frozen tissue samples consistently displayed an up-regulation of CA2 in all glioblastomas at recurrence (Figure 5D). This increase was highly significant (*p* < 0.001) in seven out of eight pairs. Averaging of all eight cases leads to a 9.1-fold up-regulation of CA2 in recurrent tumors.

This finding was corroborated by IHC staining of CA2 in FFPE tissue sections from the same eight patient matched primary and recurrent tumors. Positive staining for CA2 was observed in the cytoplasm as well as in the nuclei of cells (Figure 5E). Staining was quantified using QuPath and Fiji Image J and results are presented as the percentage of CA2 positive area in respect to the total area of the tissue sections (Figure 5E). The positively stained area varied between 3 and 38% in different slides. Comparable to our data from mRNA level (RNAseq and qPCR), the protein expression was up-regulated in recurrent tumors in the majority of the analyzed pairs. In six out of eight pairs an up-regulation of CA2 in the recurrent tumor was observed, whereas in the remaining two pairs a down-regulation was displayed. Since the immunohistochemical staining was conducted on one slide per patient only we performed no statistical evaluation. Both methods, qPCR and IHC, show a trend of up-regulation in recurrent GBMs compared to their matched primary tumors. The discrepancy between the two methods (see patient 1 and 5) is most likely due to local effects of the heterogeneous tumor landscape as the fresh frozen material which was used for qPCR did not necessarily originate from the same region as the FFPE sample used for IHC.

### 2.6. Inhibition of CA2 Leads to Resensitization of TMZ Resistant Cells

To prove a causal relationship between CA2 overexpression and TMZ resistance, CA2 was inhibited in TMZ resistant GSCs. To this end, cells were treated with IC_50_ values of TMZ, to avoid high amounts of toxic solvent, either alone or in combination with 100 µM ACZ. As expected treatment with TMZ alone led to an approximately 50% reduction in viability compared to untreated cells, which was significant in all three cases (*p* < 0.01), whereas ACZ alone did not have any effect on viability. Most remarkably, co-treatment led to a significantly more pronounced reduction in viability than single treatment with TMZ only (Figure 6). This effect was more excessively observed in TMZ resistant cells than in DMSO control cells, this could be due to the fact that CA2 is overexpressed in TMZ resistant cells as previously shown (Figure 5A to C). As ACZ itself showed low toxicity at the used concentration of 100 µM, a pilot study with higher concentrations was conducted. However, higher ACZ concentrations did not show a more pronounced effect when combined with TMZ (Appendix A). In conclusion, we show here that by co-treatment with CA2 inhibitor ACZ and TMZ a resensitization of TMZ resistant cells can be achieved.

## 3. Discussion

Formation of tumor recurrence can be considered to be a Darwinian process. While treatment eliminates most of the malignant cells, it simultaneously selects for resistant ones [12] preventing repeated and efficient treatment with the same drug. In the case of TMZ, a significant prolonged median survival of patients treated with radiotherapy and TMZ (14.6 months) was observed in comparison to patients treated only with radiotherapy (12.1 months) by Stupp et al. in 2005 [2]. The two-year survival even increased from 10.4% of patients treated only with radiation compared to 26.5% of those who received TMZ in addition [2], justifying the establishment of chemotherapy with TMZ into the guidelines for GBM treatment. However, many studies have demonstrated that TMZ also promotes malignant transformation and acquisition of resistance by hypermutation [10,11,12,13,14,15]. Addressing the former, Thunijl et al showed the capacity of TMZ to induce progression of low-grade gliomas (LGG) to GBMs by evoking mutagenesis. GBMs arising from TMZ treated LGG exhibited a 39- to 133-fold increase in mutation rate compared to their initial paired LGG [14]. Similar evidence of hypermutation was obtained when analyzing matched primary and recurrent GBMs. About 17% of recurrent tumors from patients previously treated with TMZ showed hypermutation (defined here as >500 mutations) and harbored about 10-fold as many somatic mutations as untreated tumor samples. In contrast, none of the recurrent tumors of patients who did not receive any TMZ treatment displayed hypermutation [12].

As well documented as the surge of mutation rate caused by TMZ is, there is a major lack of reports defining common pathways in vitro and in vivo which might be responsible for resistance. To achieve this, we generated TMZ resistant GSCs cells by continuous long-term exposure to TMZ and subjected them to RNA sequencing analysis. 49 genes exhibited differential expression (adjusted *p*-value < 0.1 and |log2-fold change| > 0.5) between TMZ resistant and DMSO control cells. However, it is interesting to note that CA2 is the only gene consistently up-regulated in TMZ resistance in vitro and in recurrent GBM samples. Subsequent validation by qPCR qualified CA2 to be investigated in more detail. As a result, of this, CA2 was found to be up-regulated on mRNA as well as on protein level in TMZ resistant compared to DMSO control cells. Importantly, this increase was confirmed in patient matched tissue samples from primary and recurrent GBMs of patients who were treated with TMZ identifying CA2 up-regulation as a potential mechanism of TMZ resistance.

The family of carbonic anhydrases (CAs) consists of 13 zinc containing metalloenzymes which catalyze the reversible hydration of carbon dioxide to a bicarbonate ion and a proton (Figure 7) [16,17,18].

As such CAs play an important role in several physiological processes including pH homeostasis, regulation of glycolysis and gluconeogenesis, and CO_2_ transport [17]. Furthermore, several members including CA2, 9 and 12 have been associated with neoplastic growth [16,18]. As a highly active cytosolic isoform, CA2 is expressed in almost all tissues throughout the body including the brain [16,17,18]. It has been investigated in several tumor entities such as leukemia, melanoma, neuroectodermal tumors, medulloblastomas and gliomas [16,19]. In gliomas, CA2 expression not only seems to correlate with malignancy but also with survival, suggesting a link between high CA2 expression levels and a shorter overall survival [19]. Moreover, several studies show a reduction of invasiveness by inhibition of CAs [16,19]. Sulfonamides are the most commonly used inhibitors of CAs which find clinical application in the treatment of glaucoma, epilepsy, congestive heart failure, mountain sickness and gastric as well as duodenal ulcers [20]. One such sulfonamide is acetazolamide (ACZ) which has previously been reported to have synergistic effects when combined with TMZ [21,22,23]. ACZ in combination with TMZ was shown to decrease cell viability and increase apoptosis in GBM cells in a more distinct manner than TMZ alone [21,22,23]. In established human GBM cell lines an increased apoptosis rate was due to up-regulation of Bax as well as Caspase 9 activity and a concomitant down-regulation of Bcl-2 [21] indicating that ACZ treatment interferes with anti-apoptotic mechanisms. Indeed, CA2 was identified as a Bcl-3 target gene [23]. Consequently, an increase in TMZ induced cytotoxicity was observed upon knockdown of CA2, whereas a reduction of TMZ caused an induced cell death when CA2 was overexpressed. Moreover, the authors demonstrate a chemosensitizing effect of ACZ in vitro as well as in vivo using intracranial xenografts [23]. This favors a combinatorial therapy of ACZ with TMZ, an ongoing phase I clinical trial [24] highlighting the promising therapeutic potential of ACZ.

While these studies suggest a therapeutic benefit of CA2 inhibition by ACZ for untreated GBM, none of them examined the effect on TMZ resistant GBMs. To our knowledge, this is the first study to substantiate the value of CA2 inhibition for TMZ resistant GBM recurrence. We show here a synergistic effect of ACZ and TMZ in TMZ resistant GBM stem-like cells. It is tempting to speculate that the pH regulating function of CA2 might be involved in the chemosensitizing effect of ACZ. 

The pH value is defined as the decimal logarithm of the reciprocal of the proton activity [25]. Demonstrated by the reaction outlined above the concentration of protons depends on the activity of CA2 among other factors. Maintenance of intracellular pH values is essential for basic cellular functions including enzyme activity, energy metabolism and posttranslational modification of proteins [25]. Apart from physiological processes, the efficacy of pharmaceutical agents can also depend on pH levels, as is the case for TMZ.

TMZ as prodrug spontaneously converts into the metabolite 5-(3-methyltriazen-1-en-1-yl)-1H-imidazole- 4-carboxamide (MTIC). MTIC is then transformed further into the inactive 4-amino-1H-imidazole-4-carboxamide (AIC) and a methyldiazonium ion which actually mediates the cytotoxic effect by transfer of its methyl group onto the O^6^-position of guanine. The two steps of this reaction depend on pH, in particular TMZ bioactive conversion most efficiently at physiological pH (Figure 8) [26].

Thus, further studies analyzing pH values as well as hypoxic features influencing TMZ treatment either alone or in combination with ACZ will be necessary to mechanistically explain the ACZ driven chemosensitization.

We demonstrate a causal relationship between TMZ resistance in primary stem-like cells and CA2 up-regulation. In line with previous studies mentioned above, co-treatment of ACZ and TMZ led to a significant re-sensitization of cells to TMZ, revealing a therapeutic benefit of CA2 inhibition. Mechanistically, these therapeutic benefits might be due to a subsequent intracellular acidification [20,27]. As mentioned before, the highest TMZ efficacy is achieved at physiological pH values. However, tumor cells often exhibit an intracellular alkaline pH, so that acidification as a result of CA2 inhibition causes a shift towards a physiological pH value. This is due to their altered metabolism as a consequence of enhanced glycolysis, which results in accumulation of protons and acids such as lactate. Contrary to expectations this does not lead to intracellular but rather to extracellular acidification [25,26,28,29]. It is, therefore, likely that the ACZ dosage of 100 mM causes a “therapeutic window” in which the intra- and extracellular pH values are optimal for increased TMZ efficacy. Moreover, the described alterations of the metabolism in vivo often correlate with hypoxia in a causal manner [22,23]. Hypoxia is a characteristic feature for the GBM microenvironment and impacts several processes which leads to progression of tumors such as differentiation, invasion and angiogenesis [28]. Further studies analyzing pH values as well as hypoxic features influencing TMZ treatment either alone or in combination with ACZ will be required to explain the mechanistic meaning of our results. These revealed hypoxia as one of five hallmark gene sets significantly enriched in TMZ resistant compared to DMSO control cells. In this respect, it is interesting to note that TMZ resistant GSCs selected can recapitulate gene expression changes relevant for GBM recurrence.

In conclusion, we show here for the first time to our knowledge that re-sensitization of TMZ resistant GSCs by inhibition of CA2 bears a promising therapeutic potential to overcome chemotherapeutic resistance and potential marker to assess TMZ sensitivity.

## 4. Materials and Methods

### 4.1. Isolation of Primary GBM Stem-Like Cells

We obtained approval from the Ethics committee of the Faculty of Medicine, Philipps University Marburg (institutional review board number 185/11), to collect tumor tissue samples from patients who underwent surgical resection of GBM after giving written informed consent. The tumor tissue was mechanically minced and enzymatically digested with accutase for 30 min at 37 °C. The supernatant was then passed through a 40 µm cell strainer to remove tissue fragments. Erythrocytes were lysed by incubating in Red Cell Lysis Buffer for 10 min. Cells were then seeded on a 6 well coated with 20 µg/mL laminin in PBS.

### 4.2. Cultivation of Primary GBM Stem-Like Cells

The established human GBM cell line U87 was cultured in DMEM medium containing 10% FCS, Penicillin/Streptomycin (0.1 mg/mL), Non-essential amino acids (1×), Sodium Pyruvate (1 mM). The patient-derived GBM stem-like cells were grown in DMEM/F12 (GlutaMAX) containing 2% B27 Supplement, 1% Amphothericin, 0.5% HEPES and 0.1% Gentamycin with addition of EGF and bFGF in a final concentration of 0.02 ng/µL in non-cell-culture-treated petri dishes, where they formed spheres. All cells were grown in a humidified atmosphere at 37 °C under 5% CO_2_.

### 4.3. Differentiation Assay

300,000 cells were seeded in 6-wells and cultured in 3 different conditions for 7 days: complete medium + 0.02 ng/µL EGF and bFGFcomplete medium + 1% FCSDMEM/F12 base medium + 10% FCS

After 7 days RNA was extracted and cells were analyzed for the mRNA expression of stem cell marker CD133 and differentiation marker GFAP.

### 4.4. Invasion Assay

The invasion assay was executed as described previously [30]. Briefly, the percentage of the 25,000 seeded cells which invaded along an FCS gradient through the 8 µm pores of the transwell membrane into the matrigel was quantified.

### 4.5. Generation of TMZ Resistant Cells

To generate TMZ resistant cells, they were exposed to a daily dose of 10 µM TMZ on 5 out of 7 weekdays for a period of at least 20 weeks. For comparison an untreated control as well as a solvent (DMSO) control were also done. Resistance was validated and quantified by Dose-Response-Curves based on viability assays.

### 4.6. Viability Assay

96 well plated were coated with 50 µg/mL collagen (Sigma-Aldrich, Munich, Germany) in 0.01 M HCl and 500 cells/well were seeded. After 24 h the indicated concentrations of TMZ and/or ACZ were administered to wells in triplicates. Cells were cultured for 2 weeks before viability was measured. For this, 50 µL CellTiter-Glo^®^ 3D Cell Viability Assay (Promega, Fitchburg, MA, USA) was added directly to the well, followed by 15 min of vigorous shaking and 15 min incubation at room temperature in the dark. Luminescence was determined using a 96 well plate reader (FLUOstar OPTIMA Microplate Reader, BMG LABTECH, Ortenberg, Germany). Data was normalized to untreated cells. The IC_50_ values were determined by graphical analysis of the plotted data.

### 4.7. Side Population Analysis

5 µg Hoechst 33342 dye was added to 1 million cells in 1 mL medium and incubated for 90 min at 37 °C with or without 50 µM Verapamil (Sigma-Aldrich, Munich, Germany). After this time cells were constantly kept on ice to inhibit further efflux of the dye. After centrifugation and washing, 2 µg/mL PI was added and cells were analyzed at the FACS Analyse MoFlo^TM^ Astrios High End Sorter (Beckman Coulter GmbH, Krefeld, Germany). By gating properly debris, doublets and dead cells were excluded before measuring Hoechst intensity at two wavelengths (355–488 and 355–620).

### 4.8. RNA Extraction and qPCR

To extract RNA, 50 mg tumor tissue was mechanically homogenized in 1 mL Qiazol (Qiagen, Hilden, Germany), whereas cells were lysed in 1 mL Qiazol by resuspending. The following steps were performed as previously described [30], including transcription to cDNA and qPCR.

### 4.9. RNA Sequencing, Subsequent Data Analysis, Differential Expression Analysis, and Gene Set Enrichment Analysis (GSEA)

RNA for Sequencing was isolated using the Macherey-Nagel NucleoSpin RNA Plus Kit according to manufactures’ instructions. Integrity of total RNA was assessed on the Bio-Rad Experion. Sequencing libraries were prepared with the TruSeq Stranded mRNA Kit (Illumina, San Diego, CA, USA). On-board cluster generation using the TruSeq Rapid SR Cluster Kit-HS (Illumina) and single read 50 nucleotide sequencing was performed on a HiSeq Rapid SR Flow Cell (Illumina) on the Illumina 1500 platform.

Transcript quantification was carried using the *Salmon* approach and the transcript reference from the GRCH38 annotation [31]. Data import into R and gene expression quantification was realized with *tximport* R package [32]. Differential gene expression analysis between TMZ resistant cell group (all three cell lines) and DMSO control cell group (all three cell lines) was performed with a paired design (TMZ resistant vs. DMSO control), using the methods implemented in the *DESeq2* (Version 1.14.1, default settings) R package, wherein genes with an absolute log2-fold change > 0.5 and an adjusted *p*-value < 0.1 were considered differentially expressed [33]. Gene Set Enrichment Analysis (GSEA) was conducted in pre-ranked mode, using the log2-fold change values obtained from *DESeq2* for the ranking of the gene list and the *fgsea* R package [34,35]. The Hallmark, (GO biological process, and Reactome gene sets) were downloaded from Molecular Signature Database [36] and used for enrichment testing. Gene sets with an adjusted *p*-value <0.05 were considered significantly enriched.

### 4.10. Protein Extraction and Western Blot

Protein extraction, sample preparation and western blotting were conducted as previously described [30]. The following primary antibodies were used: anti-CA2, Abcam ab124687, 1:1000 in 5% milk in TBST and anti-β-Tubulin, Novus Biologicals, NB600-936, 1:2000 in 5% milk in TBST.

### 4.11. Immunohistochemistry

Formalin fixed and paraffin embedded tissue sections (3 µm) were stained using the VECTA Stain Elite Kit (Vecta Laboratories, Burlingame, CA, USA) according to manufacturer’s instructions. After deparaffination at 60 °C for 45 min, sections were hydrated using descending alcohol concentrations. Demasking of epitops was achieved by boiling in citratbuffer (10 mM Trisodium citrate dihydrate, pH = 6). Endogenous peroxidase was blocked by incubating in 3% H_2_O_2_ in methanol for 30 min before incubation in 1.5% goat serum for blocking of unspecific binding. Followed by incubation with primary antibody (anti-CA2, Abcam ab124687, 1:250 in PBS) at 4 °C overnight. Incubation with the respective secondary biotinylated antibody and ABC reagent were followed by DAB staining with the ImmPACTT DAB Kit (Vecta Laboratories, Burlingame, CA, USA). For counterstaining hematoxylin (Carl-Roth, Karlsruhe, Germany) was used. Finally, sections were dehydrated by ascending alcohol concentrations and covered with mounting medium. Images of whole sections were acquired with Axio Scan.Z1 (Zeiss, Jena, Germany) and processed using QuPath [37] and Fiji ImageJ [38].

### 4.12. Stereotactic Injection and In Vivo Analysis of Tumor Growth

For in vivo experiments 10 to 12-week-old athymic nude mice were obtained from Harlan, Indianapolis, USA. Mice were anesthetized using 1–3% isoflurane and injected with 100,000 cells in 10 µL PBS into the corpus striatum using a stereotactic device. Tumor growth was monitored for 2 to 8 weeks post injection using a 7T MRI (Clinscan 70/30 USR Bruker). When abortion criteria were reached, animals were euthanized in accordance with the local guidelines and the whole brain was immediately collected for histological analysis. Mouse brains were fixed overnight in 4% formalin and embedded in paraffin. Paraffin sections were stained as described above.

Animal facilities and experiments were authorized by the Regierungspräsidium Gießen Germany, according to the German and Hessen animal welfare regulations (file number G60-2016). Animals were housed in the special pathogen-free facility, where a constant temperature of 26 °C, a 12 h light–12 h dark electric cycle, water ad libitum and a commercial laboratory animal diet were provided.

### 4.13. Statistical Analyses

Data were analyzed using the statistical software R. If not indicated otherwise, data are presented as mean values ± SD. Student’s *t*-tests were employed to determine significance, which was indicated with one, two or three stars for *p*-values of <0.05; <0.01 and <0.001 respectively.

## 5. Conclusions

As recurrence caused by TMZ resistance is almost unavoidable for GBM patients, there is an urgent need for new treatment options. Here we show the potential of CA2 as a new potential target, as it is up-regulated in patient matched GBM tissue samples of primary and recurrent GBMs as well as in TMZ resistant primary GBM stem-like cells, where its inhibition by ACZ led to a resensitization of resistant cells.

## Figures and Tables

**Figure 1 cancers-11-00921-f001:**
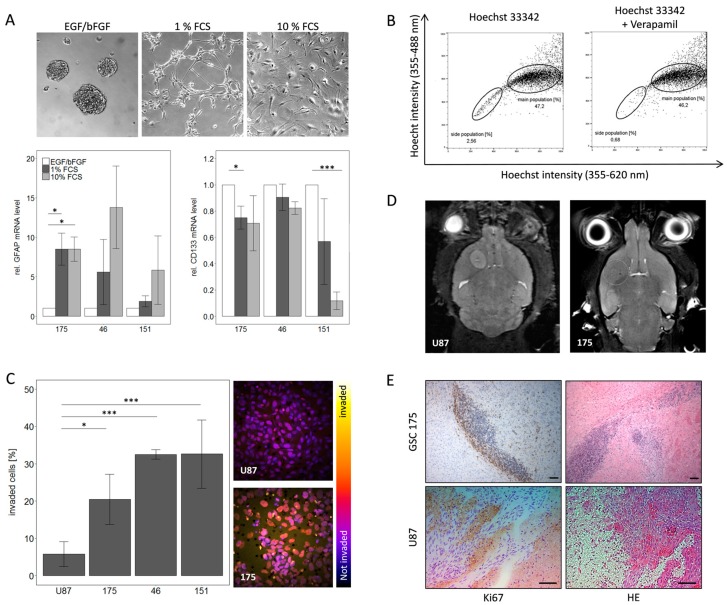
Characterization of primary GBM stem-like cells regarding their stem cell phenotype and GBM specific features. (**A**) Differentiation of primary GSCs can be induced by FCS leading to a phenotypic change as well as alterations in gene expression for the stem cell marker CD133 and the differentiation marker GFAP. (**B**) Further validation of stem-like character was achieved by side population analysis which showed a specific population with a higher efflux of Hoechst 33342, this effect is due to higher activity of ABC transporters since it can be blocked with verapamil. (**C**) Invasiveness of GSCs is much higher compared to the established cell line U87 in vitro. (**D**) MRI analysis of mice and (**E**) HE and Ki67 staining of tumors derived from GSC 175 and U87 cells in vivo. Please note that Ki67 positive cells can be detected in the tissue surrounding the tumor mass after injection of GSC175 cells, but not after injection of U87 cells. Scale bar: 100 µm, respectively.

**Figure 2 cancers-11-00921-f002:**
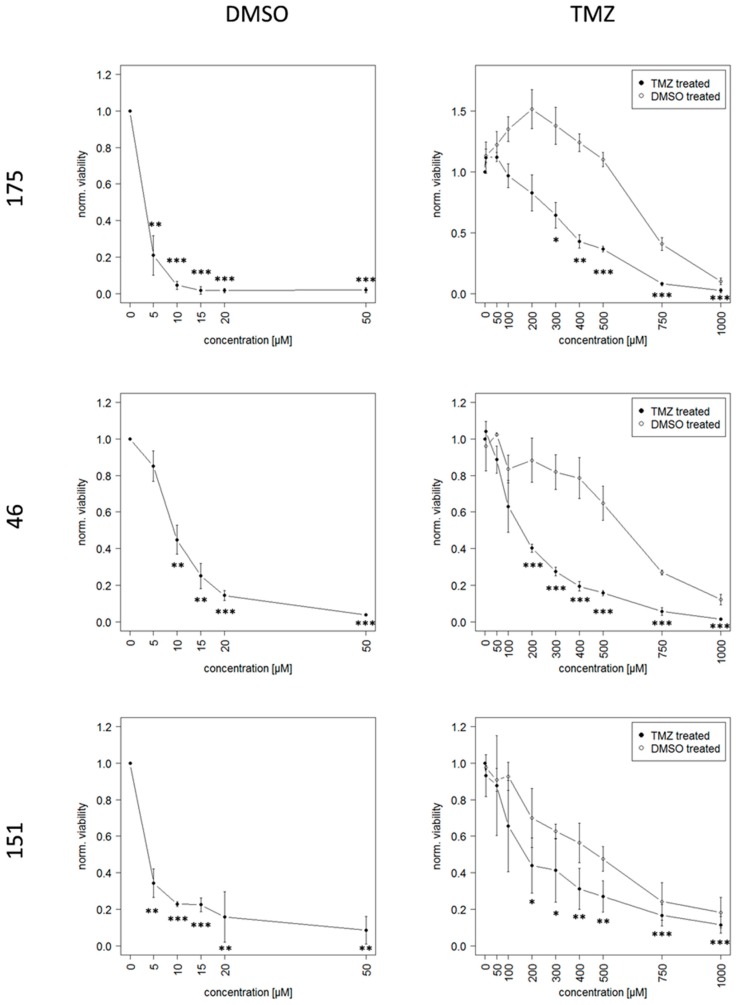
Dose-Response-Curves of TMZ resistant and DMSO control cells based on the viability measurement with CellTiter-Glo^®^ 3D Cell Viability Assay (Promega). Please note that the *X*-axis scale has different dimensions, the half lethal dose is between 15 and 100-fold higher in cells that previously were continuously exposed to TMZ and thereby acquired resistance to TMZ. Significance is indicated for each concentration compared to the respective untreated cells with * *p* < 0.05; ** *p* < 0.01; *** *p* < 0.001.

**Figure 3 cancers-11-00921-f003:**
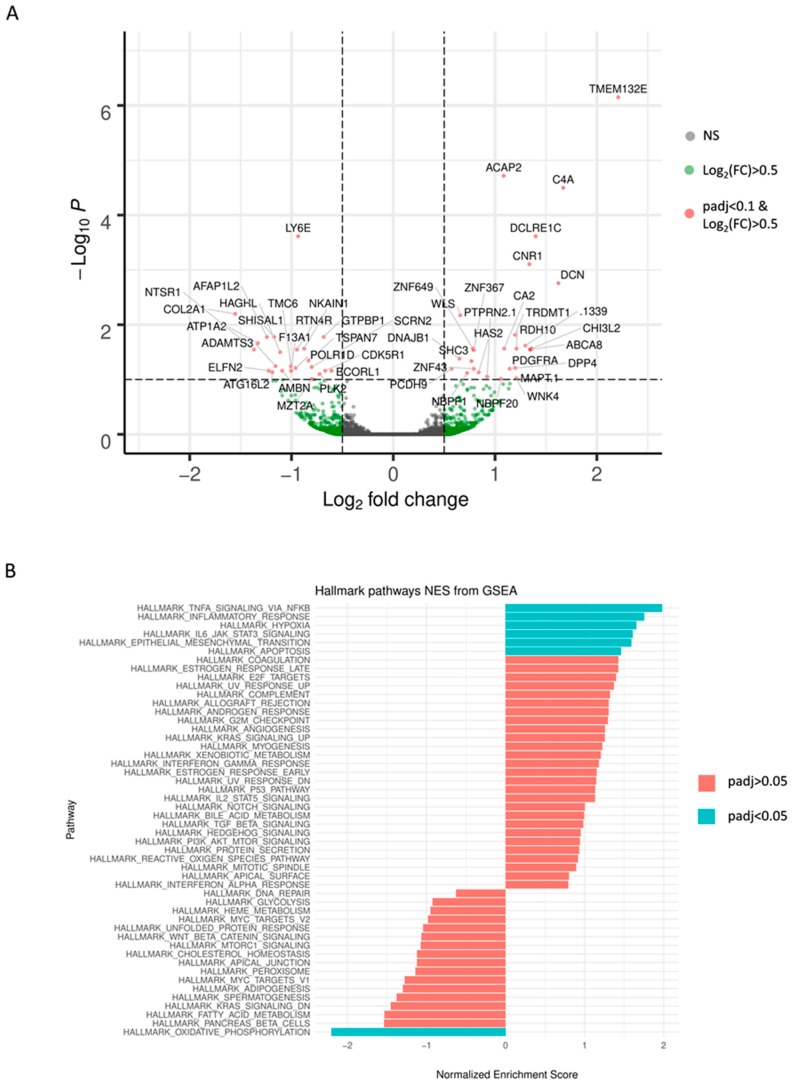
Results of the RNA Sequencing analysis are demonstrated as (**A**) volcano plot of differential expression analysis results by *DEseq2*, plotting the log2(fold change) versus the –log10(adjusted *p*-value). 49 differentially expressed genes between TMZ resistant cell group and DMSO control cell group are exceeding the threshold set at adjusted *p*-value < 0.1 and |log2-fold change| > 0.5 (red dots). (**B**) Normalized enrichment scores (NES) derived from the gene set enrichment analysis (GSEA) and ranking of the genes according to fold-changes derived by *DESeq2* analysis using the Hallmark gene sets. Only those gene sets displayed in turquoise are depleted in TMZ resistant cells with an adjusted *p*-value < 0.05. Positive NES values suggest an up-regulation of the respective set, whereas negative values indicate a down-regulation.

**Figure 4 cancers-11-00921-f004:**
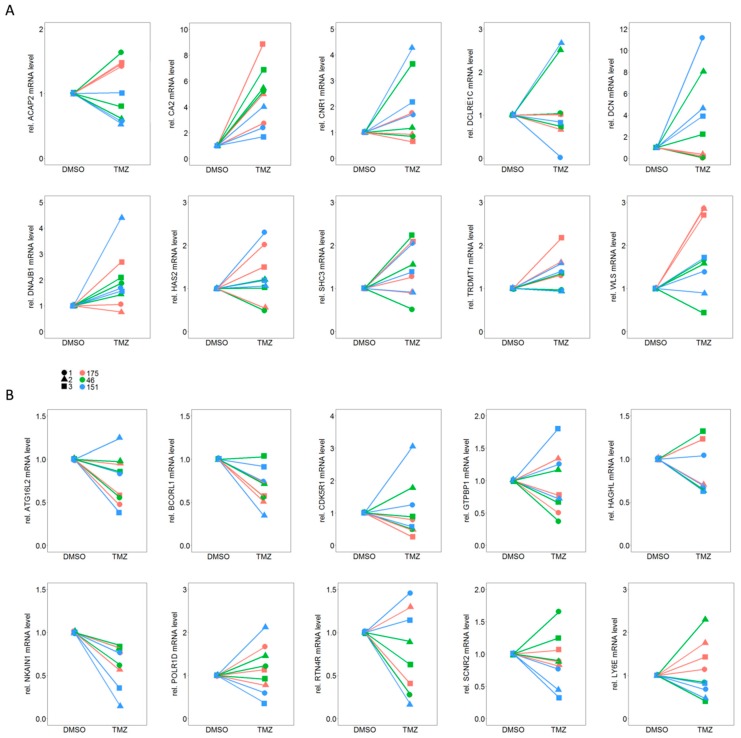
qPCR analysis for (**A**) 10 up-regulated and (**B**) 10 down-regulated genes out of the total of 49 genes which showed differential expression between TMZ resistant and their matched DMSO control cells. Please note that only for one gene from each set (A: CA2 and B: NKAIN1) the trend in regulation could be reproduced for all three independent experiments (1–3) and all three individual TMZ-DMSO pairs (175, 46, 151).

**Figure 5 cancers-11-00921-f005:**
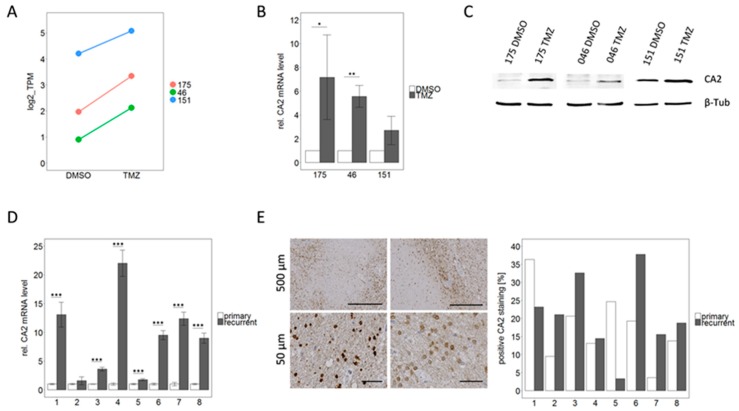
CA2 up-regulation correlates with TMZ resistance, not only in GSCs but also in patient matched samples from primary and recurrent GBMs. (**A**) CA2 was among the differentially expressed genes in RNA sequencing analysis which showed similar trends for all three TMZ-DMSO pairs. (**B**) Validation by qPCR analysis (*n* = 3). Significance levels were: * *p* < 0.05; ** *p* < 0.01 (**C**) Consistent with qPCR data, an up-regulation of CA2 in TMZ resistant cells compared to the DMSO control cells was observed by Western Blot, detailed information can be found in Appendix A. (**D**) qPCR analysis of patient matched primary and recurrent GBM tissue reinforces the impression that CA2 up-regulation is linked to TMZ resistance, since an up-regulation of CA2 was observed in recurrent tumor tissue compared to the respective primary tumor tissue. This effect was significant in 7 out of 8 pairs for *n* = 3 independent experiments. *** *p* < 0.001 (**E**) IHC for CA2 performed on FFPE samples from the same patients showed immunoreactive tumor cells as exemplified by pictures on the left. On the right quantification is visualized as the percentage of positively stained area in respect to the total area of the sections.

**Figure 6 cancers-11-00921-f006:**
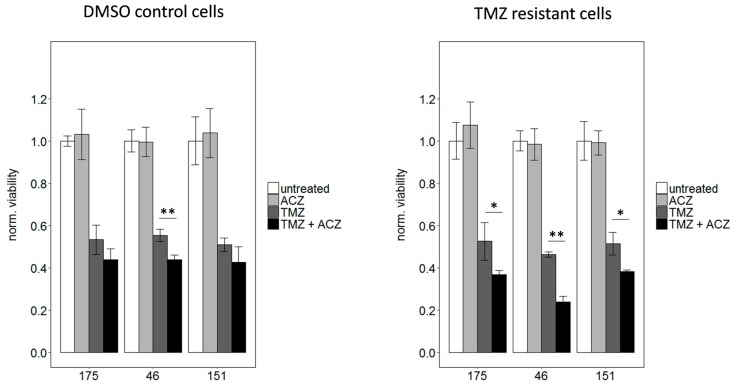
Co-treatment with the respective IC_50_ dosages of TMZ and 100 µM ACZ decreased viability significantly compared to treatment with TMZ alone. This effect is less pronounced for DMSO control cells shown on the left with the used IC_50_ values of 3 µM for 175, 7.5 µM for 46 and 2.5 µM for 151. For TMZ resistant cells IC_50_ values of 250 µM for 175, 150 µM for 46 and 250 µM for 151 were used. For reasons of clarity, significance was only indicated for the comparison between TMZ alone and TMZ in combination with ACZ. However, co-treatment as well as TMZ alone led to a significant reduction of viability compared to untreated cells, whereas ACZ alone had no significant impact. Treatment with equivalent amounts of DMSO led to a reduction in viability of less than 10% in comparison to untreated cells. Significance determined: * *p* < 0.05; ** *p* < 0.01.

**Figure 7 cancers-11-00921-f007:**
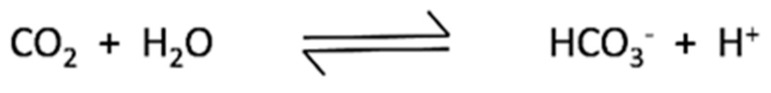
Reaction catalyzed by Carbonic anhydrases.

**Figure 8 cancers-11-00921-f008:**
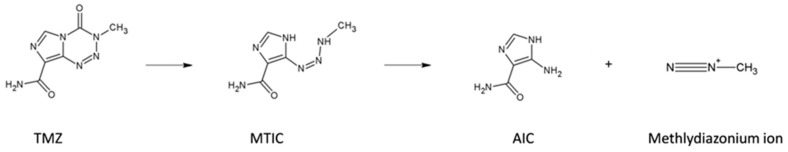
Reaction scheme of TMZ activation.

**Table 1 cancers-11-00921-t001:** Clinical data of patient donors for GSCs. MGMT Status: promotor methylation status of the O^6^-methylguanine-DNA-methyltransferase, IDH R132H: wildtype, no detectable R132H point mutation of isocitrate dehydrogenase 1 (IDH1) in the tumor tissue.

No.	Sex	Age	MGMT Status	IDH1 R132H
175	m	65	methylated	wildtype
46	f	43	methylated	wildtype
151	m	66	methylated	wildtype

**Table 2 cancers-11-00921-t002:** Percentage of DMSO solvent for each respective TMZ concentration used for establishing the Dose-Response-Curves shown in Figure 2.

Concentration (µM)	Amount DMSO (%)
5	0.0097
10	0.019
15	0.029
20	0.039
50	0.097
100	0.19
200	0.39
300	0.58
400	0.78
500	0.97
750	1.46
1000	1.94

**Table 3 cancers-11-00921-t003:** Data describing the patient cohort used for patient matched analysis of first manifested and recurrent tumors, including gender, age at time of initial diagnosis, survival and latency given in days. Furthermore, histopathologic data such as promotor methylation status of the O^6^-methylguanine-DNA-methyltransferase (MGMT), expression of epidermal growth factor receptor (EGFR) variant III, the existence of the R132H point mutation of isocitrate dehydrogenase (IDH) and the Ki67 Labeling index (Ki67Li) are given for first manifested and recurrent tumor individually.

Pair	Sex	Age	Survival	Latency	MGMT Status	EGFR v III	IDH1 R132H	Ki67
1	m	51	371	172	not methylated	positive	negative	25%
					not methylated	negative	negative	10%
2	m	52	760	491	not methylated	positive	negative	75%
					not methylated	negative	negative	20%
3	f	51	445	147	not methylated	negative	negative	20%
					not methylated	negative	negative	5%
4	m	57	343	288	not methylated	negative	negative	up to 50%
					not methylated	negative	negative	10%
5	f	77	550	266	not methylated	negative	negative	15%
					not methylated	positive	negative	7%
6	m	48	524	603	not methylated	negative	negative	25%
					not methylated	negative	negative	up to 50%
7	m	71	532	357	Methylated	positive	negative	50%
					not methylated	negative	negative	50%
8	m	45	576	397	not methylated	negative	negative	30%
					not methylated	negative	negative	up to 5%

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
