# Peer review of "Comparative Transcriptomic Analysis of Temozolomide Resistant Primary GBM Stem-Like Cells and Recurrent GBM Identifies Up-Regulation of the Carbonic Anhydrase CA2 Gene as Resistance Factor"

_cancers, 2019, doi:10.3390/cancers11070921_

Round 1
Reviewer 1 Report
In this interesting work, the authors compared the gene expression between temozolomide (TMZ) resistant GBM stem-like cells (GSCs) and their matched DMSO control cells. Subsequently, they identified a total of 49 differentially expressed genes, among which carbonic anhydrase 2 (CA2) well correlate with the glioma malignancy and patient survival. Finally, they demonstrated that the co-treatment with the CA2 inhibitor acetazolamid (ACZ) sensitized cells to TMZ induced cell death. The manuscript is overall well written. A few commons for the authors’ consideration:
1. Fig. 1E, the authors claimed that the primary 175 cells showed higher invasive behavior compared to U87 cells, as evidenced by the invasion along the corpus callosum away from the point of injection. I wonder whether they also did the Ki67 and HE staining for U87-derived tumors for comparison.
2. Fig. 6, why was the 100 µM was selected for ACZ treatment?
3. Page 9, line 216, higher ACZ concentrations did not show a more pronounced effect (data not shown); however, I think it would be more informative if these data are shown in the revised manuscript.
4. Fig. 6, the DMSO control GSCs should also used for a comparison to the TMZ resistant cells. It would be interesting to check whether co-treatment with ACZ and TMZ also decreases viability in DMSO control cells compared to the TMZ treatment alone.
5. Page 12, line 305, it is speculated that the therapeutic benefits of ACZ might be caused by the intracellular acidification. Higher concentrations of ACZ could provide more acidification, which may not be able to further increase the TMZ efficacy, therefore did not show a more pronounced effect. More discussion for this interesting point is encouraged.
Author Response
Reviewer #1:
In this interesting work, the authors compared the gene expression between temozolomide (TMZ) resistant GBM stem-like cells (GSCs) and their matched DMSO control cells. Subsequently, they identified a total of 49 differentially expressed genes, among which carbonic anhydrase 2 (CA2) well correlate with the glioma malignancy and patient survival. Finally, they demonstrated that the co-treatment with the CA2 inhibitor acetazolamid (ACZ) sensitized cells to TMZ induced cell death. The manuscript is overall well written. A few commons for the authors’ consideration:
1. Fig. 1E, the authors claimed that the primary 175 cells showed higher invasive behavior compared to U87 cells, as evidenced by the invasion along the corpus callosum away from the point of injection. I wonder whether they also did the Ki67 and HE staining for U87-derived tumors for comparison.
Authors’ Response: We have now included HE and Ki67 staining for a representative U87-derived tumor in Fig. 1E.
2. Fig. 6, why was the 100 µM was selected for ACZ treatment?
Authors’ Response: Initially, a pilot study was conducted to determine the most suitable ACZ concentration. Interestingly, higher concentrations of ACZ did not show a more pronounced effect. In order to keep toxic effect of the solvent (DMSO) at a minimum, the lowest effective dose (100 µM) was used in our co-treatment assays. Notably, this concentration was also used in most of the studies cited in our manuscript [21-23].
3. Page 9, line 216, higher ACZ concentrations did not show a more pronounced effect (data not shown); however, I think it would be more informative if these data are shown in the revised manuscript.
Authors’ Response: We have now included the data from the pilot study in the supplement.
4. Fig. 6, the DMSO control GSCs should also used for a comparison to the TMZ resistant cells. It would be interesting to check whether co-treatment with ACZ and TMZ also decreases viability in DMSO control cells compared to the TMZ treatment alone.
Authors’ Response: This was actually done but data was not included in the manuscript, as the effect of ACZ on naïve GBM cells has been shown in several publications (see references 21 to 23) and we wanted to emphasize the novelty of our approach using TMZ resistant cells. However, we included these data in Fig. 6 now.
5. Page 12, line 305, it is speculated that the therapeutic benefits of ACZ might be caused by the intracellular acidification. Higher concentrations of ACZ could provide more acidification, which may not be able to further increase the TMZ efficacy, therefore did not show a more pronounced effect. More discussion for this interesting point is encouraged.
Authors’ Response: This is indeed an interesting observation. Our current view is that the dosage of ACZ determines a pH optimum that serves as a “therapeutic window” for the increased efficacy of TMZ. We have added a corresponding statement in the discussion on page 12, line 330-332.
Reviewer 2 Report
What is the reason for demonstrating a MDR-expressing, verapamil-sensitive subpopulation in the primary cells in Fig 1B? Is there significance to the 2.56% of cells with respect to the later experiments with TMZ-acquired resistance? It would be helpful for the reader if this is clarified in the manuscript.
How much DMSO is being added as vehicle in Fig 2, TMZ-treated samples? Does the amount of final DMSO in cell culture increase from left to right? If so, the % vol (or concentration) of DMSO should be included as values on the graph, or shown as a separate axis. It would also be preferable to avoid using cytotoxic levels of DMSO, if possible.
Approximate IC50 values for TMZ against lines 175, 46, and 151 are reported in the legend of Fig 6. How were the IC50 values determined, by visual inspection? Would suggest calculating IC50 from viability data and reporting those values with confidence intervals as part of Fig 2 (as a more accurate measure of IC50).
It is not clear what whether the reported gene expression changes from DEseq2 analysis (e.g. Volcano plot in Fig 4) represent average values for each condition (DMSO or TMZ) from the three cell lines, or individual values for a single representative cell line? Please include more description in the text of the manuscript, and in the figure legend, so this is clear to the reader.
Why is there little reproducibility in the qPCR data of Fig 4? It appears that out of 49 changed genes, only two consistently change in the same direction between the three experimental replicates. This is acknowledged in the manuscript, but should be more thoroughly addressed, since it brings into question the reliability of the RNA seq results.
For Fig 6, it would be more reliable to test a range of TMZ concentrations +/- ACZ and see how much it left-shifts the concentration-response curve (IC50) for TMZ.
Suggest using gene-silencing or genetic-disruption of CA2 as an alternative to ACZ treatment, to test the significance of CA2 in acquired resistance to TMZ. This would be a more refined approach that would test the specific contribution of CA2 (and avoid confounding effects of ACZ, such as inhibition of other CA isoforms).
Author Response
Reviewer #2:
1. What is the reason for demonstrating a MDR-expressing, verapamil-sensitive subpopulation in the primary cells in Fig 1B? Is there significance to the 2.56% of cells with respect to the later experiments with TMZ-acquired resistance? It would be helpful for the reader if this is clarified in the manuscript.
Authors’ Response: Similar to the differentiation assay, the side population assay was conducted to demonstrate the stem cell character of the GSCs generated. The side population assay is frequently used in the literature and it is standard practice to determine specificity of the side population by blocking efflux of the Hoechst dye by inhibiting transporters, i.e. with verapamil. The side population assay simply served as a proof of concept, and there is no significance to the percentage of cells in the side population with regard to the later experiments.
2. How much DMSO is being added as vehicle in Fig 2, TMZ-treated samples? Does the amount of final DMSO in cell culture increase from left to right? If so, the % vol (or concentration) of DMSO should be included as values on the graph, or shown as a separate axis. It would also be preferable to avoid using cytotoxic levels of DMSO, if possible.
Authors’ Response: We apologize for not explaining this sufficiently. The amount of DMSO increases from left to right as TMZ concentrations increase. We have now included a table showing the % vol of DMSO for each respective TMZ concentration used for establishing the Dose-Response-Curves shown in Figure 2. This table shows that it was not entirely possible to avoid cytotoxic levels of DMSO due to the considerably low solubility of TMZ.
3. Approximate IC50 values for TMZ against lines 175, 46, and 151 are reported in the legend of Fig 6. How were the IC50 values determined, by visual inspection? Would suggest calculating IC50 from viability data and reporting those values with confidence intervals as part of Fig 2 (as a more accurate measure of IC50).
Authors’ Response: The IC50values provided in the legend of Fig. 6 were determined from the graphs. We have now addressed this in the Materials and Methods section. We acknowledge that this is not the most precise method, however as can be seen from Fig. 6 the determined values showed a high reproducibility. We believe that a retrospective calculation of IC50values would result in comparable values. Therefore we renounce from doing so, since the values given were also used for the further experiments.
4. It is not clear whether the reported gene expression changes from DEseq2 analysis (e.g. Volcano plot in Fig 4) represent average values for each condition (DMSO or TMZ) from the three cell lines, or individual values for a single representative cell line? Please include more description in the text of the manuscript, and in the figure legend, so this is clear to the reader.
Authors’ Response: We apologize for not explaining this sufficiently. We have now clarified this in the Material and Methods section as well as in the figure legend of Fig. 3.
5. Why is there little reproducibility in the qPCR data of Fig 4? It appears that out of 49 changed genes, only two consistently change in the same direction between the three experimental replicates. This is acknowledged in the manuscript, but should be more thoroughly addressed, since it brings into question the reliability of the RNA seq results.
Authors’ Response: We can only speculate, however the RNA sequencing was only done from n=1 technical replicate. In our opinion, having n=3 biological replicates (different cell lines) was more relevant, since it emphasizes the physiological relevance and thereby the clinical impact of targets better than having technical replicates. We then used n=3 technical replicates for validation of the differentially expressed genes leading to the observed discrepancy.
6. For Fig 6, it would be more reliable to test a range of TMZ concentrations +/- ACZ and see how much it left-shifts the concentration-response curve (IC50) for TMZ.
Authors’ Response: The IC50values were chosen in order to obtain a high cytotoxic effect to be able to see a chemosensitization effect of CA2 inhibition but simultaneously keep the cytotoxic effect of the solvent DMSO as low as possible.
7. Suggest using gene-silencing or genetic-disruption of CA2 as an alternative to ACZ treatment, to test the significance of CA2 in acquired resistance to TMZ. This would be a more refined approach that would test the specific contribution of CA2 (and avoid confounding effects of ACZ, such as inhibition of other CA isoforms).
Authors’ Response: We appreciate this suggestion and have also contemplated using siRNA. However this approach carries some experimentally difficulties: i) In our hands the transfection rate of GSCs is extremely low with less than 10 %. ii) As we have previously observed a lot of cells dying after transfection, this would possible falsify the results. iii) For the viability read out an incubation time of 14 days is required, which is a longer time than siRNA effects would last. This would require even repeated transfections. Given these problems, we conclude that this approach is not feasible in GSCs at the moment.
Reviewer 3 Report
Ricarda et al observed carbonic anhydrase 2 (CA2) as a candidate gene correlated with glioma malignancy among 49 differentially expressed genes via RNA sequencing and described that both mRNA and protein levels of CA2 are upregulated in TMZ-resistant GBM cells and clinical samples. This is very interesting but revision is required for the publication of “Cancers” as below. 1. Figure 2 requires statistics. 2. As Table 2, authors defined the status of tumors from patients. Is there any meaning among MGMT, EGFR, and IDH mutation with CA2? What about the status of p53? It is better to describe this point in the discussion section. 3. In Figure 5D and E, discuss the discrepancy of mRNA and protein level in sample 1, 2 and 5. Also, it requires the clear grouping between “primary” and “recurrent” in x-axis of these figures. 4. In Figure 6 legend, the concentration of TMZ should be described as the same of ACZ. 5. It is interesting to check whether specific inhibition of CA2 (such as siRNA) leads to cell death or to re-sensitize TMZ response. 6. Is there any change of MGMT after ACZ and TMZ treatment? 7. In Materials and Methods, Institutional Review Board (IRB) number should be described.Author Response
Reviewer #3:
Ricarda et al observed carbonic anhydrase 2 (CA2) as a candidate gene correlated with glioma malignancy among 49 differentially expressed genes via RNA sequencing and described that both mRNA and protein levels of CA2 are upregulated in TMZ-resistant GBM cells and clinical samples. This is very interesting but revision is required for the publication of “Cancers” as below.
1. Figure 2 requires statistics.
Authors’ Response: We apologize for not having included statistics in Fig. 2, we have now remedied this.
2. As Table 2, authors defined the status of tumors from patients. Is there any meaning among MGMT, EGFR, and IDH mutation with CA2? What about the status of p53? It is better to describe this point in the discussion section.
Authors’ Response: We included the routinely collected histopathologic data of patients only to describe the patient cohort. It was not our aim to correlate these parameters to our findings, as n=8 patients would be not enough to obtain statistically relevant data.
3. In Figure 5D and E, discuss the discrepancy of mRNA and protein level in sample 1, 2 and 5. Also, it requires the clear grouping between “primary” and “recurrent” in x-axis of these figures.
Authors’ Response: We believe that the discrepancy is due to local effects as GBM is a very heterogeneous tumor. The material for the two experiments did not necessarily originate from the same spot, as fresh frozen tissue was used for the qPCR analysis (Fig. 5D) and FFPE samples for IHC (Fig. 5E). We now discuss this point in line 213 ff.
4. In Figure 6 legend, the concentration of TMZ should be described as the same of ACZ.
Authors’ Response: We apologize for being unclear. We have now changed the figure description from “Co-treatment with 100 µM ACZ and IC50 (175: 250 µM, 46: 150 µM, 151: 250 µM) dosages of TMZ decreased viability significantly compared to treatment with TMZ alone” to “Co-treatment with the respective IC50 dosages of TMZ (175: 250 µM, 46: 150 µM, 151: 250 µM) and 100 µM ACZ decreased viability significantly compared to treatment with TMZ alone”.
5. It is interesting to check whether specific inhibition of CA2 (such as siRNA) leads to cell death or to re-sensitize TMZ response.
Authors’ Response: We absolutely agree with this point. For technical reasons as to why we nevertheless did not perform these experiments, please see Reviewer #2 Point 7.
6. Is there any change of MGMT after ACZ and TMZ treatment?
Authors’ Response: As this was not the aim of our study we did not check the methylation status of the used GSCs after TMZ and/or ACZ treatment. However, this is a very interesting question that has previously been addressed by other groups. The MGMT promotor methylation status is unquestionable one of the most important prognostic factors for GBM patients (next to extent of resection, age, initial performance index). However the stability of the MGMT promotor methylation status throughout the course of treatment is in question. There are studies that show changes in MGMT promotor methylation and make the argument that reducing the methylation of the MGMT promotor is a mechanism of acquiring resistance in GBM in cell culture models as well as in patient samples [B]. To the contrary, there are also studies which claim the MGMT promotor methylation status is a stable biomarker in clinical specimen [C] as well as in in vitromodels using TMZ resistant cells [D].
[A] Shabierjiang Jiapaer, Takuya Furuta, Shingo Tanaka; Tomohiro Kitabayashi and Mitsutoshi Nakada. Potential Strategies Overcoming the Temozolomide Resistance for Glioblastoma. Neurol Med Chir (Tokyo)2018, 58, 405-421.
[B] Chul-Kee Park, Ja Eun Kim, Ji Young Kim, Sang Woo Song, Jin Wook Kim, Seung Hong Choi, Tae Min Kim, Se-Hoon Lee, Il Han Kim, and Sung-Hye Park. The Changes in MGMT Promotor Methylation Status in Initial and Recurrent Glioblastomas. Transl Oncol2012, 5, 393-397.
[C] Alba A. Brandes, Enrico Franceschi, Alexandro Paccapelo, Giovanni Tallini, Dario De Biase, Claudio Ghimenton, Daniela Danieli, Elena Zunarelli, Giovanni Lanza, Enrico Maria Silini, Carmelo Sturiale, Lorenzo Volpin, Franco Servadei, Andrea Talacchi, Antonio Fioravanti, Maria Pia Foschini, Stefania Bartolini, Annalisa Pession, and Mario Ermani. Role of MGMT Methylation Status at Time of Diagnosis and Recurrence for Patients with Glioblastoma: Clinical Implications.Oncologist2017, 22, 432-427.
[D] Sang Y Lee. Temozolomide resistance in glioblastoma multiforme. Genes Dis 2016, 3, 198-210.
7. In Materials and Methods, Institutional Review Board (IRB) number should be described.
Authors’ Response: We apologize for using the wrong wording “file number”, we have now changed it to “institutional review board number”.
Round 2
Reviewer 1 Report
The manuscript has been adequately revised.
Reviewer 2 Report
Concerns have been sufficiently addressed
Reviewer 3 Report
This revised manuscript is now acceptable for the publication of "Cancers".